# Factors Associated with the Risk of Major Adverse Cardiovascular Events in Patients with Ankylosing Spondylitis: A Nationwide, Population-Based Case—Control Study

**DOI:** 10.3390/ijerph19074098

**Published:** 2022-03-30

**Authors:** Chung-Mao Kao, Jun-Sing Wang, Wei-Li Ho, Tai-Ming Ko, Hsian-Min Chen, Ching-Heng Lin, Wen-Nan Huang, Yi-Hsing Chen, Hsin-Hua Chen

**Affiliations:** 1Division of Allergy, Immunology, and Rheumatology, Department of Internal Medicine, Taichung Veterans General Hospital, Taichung 40705, Taiwan; tonykao1207@yahoo.com.tw (C.-M.K.); gtim5555@gmail.com (W.-N.H.); dr.yihsing@gmail.com (Y.-H.C.); 2Division of Endocrinology and Metabolism, Department of Medicine, Taichung Veterans General Hospital, Taichung 40705, Taiwan; jswang0819@gmail.com; 3School of Medicine, College of Medicine, National Yang Ming Chiao Tung University, Taipei 11221, Taiwan; 4College of Medicine, National Chung Hsing University, Taichung 40227, Taiwan; 5Division of Allergy, Immunology and Rheumatology, Chiayi Branch, Taichung Veterans General Hospital, Chiayi 60090, Taiwan; thatsit1011@gmail.com; 6Department of Biological Science and Technology, National Yang Ming Chiao Tung University, Hsinchu 30010, Taiwan; taiming23@gmail.com; 7Institute of Bioinformatics and Systems Biology, National Yang Ming Chiao Tung University, Hsinchu 30010, Taiwan; 8Institute of Biomedical Sciences, Academia Sinica, Taipei 11529, Taiwan; 9Department of Medical Research, Taichung Veterans General Hospital, Taichung 40705, Taiwan; hsmin@vghtc.gov.tw (H.-M.C.); epid@vghtc.gov.tw (C.-H.L.); 10Center for QUantitative Imaging in Medicine (CQUIM), Department of Medical Research, Taichung Veterans General Hospital, Taichung 40705, Taiwan; 11Department of Computer Science and Information Engineering, National United University, Miaoli 36063, Taiwan; 12Institute of Biomedical Science and Rong Hsing Research Center for Translational Medicine, National Chung Hsing University, Taichung 40227, Taiwan; 13Department of Industrial Engineering and Enterprise Information, Tunghai University, Taichung 40704, Taiwan; 14Department of Healthcare Management, National Taipei University of Nursing and Health Sciences, Taipei 11219, Taiwan; 15Department of Public Health, College of Medicine, Fu Jen Catholic University, New Taipei City 24205, Taiwan; 16Division of General Internal Medicine, Department of Internal Medicine, Taichung Veterans General Hospital, Taichung 40705, Taiwan; 17Big Data Center, National Chung Hsing University, Taichung 40227, Taiwan

**Keywords:** ankylosing spondylitis, case–control study, major adverse cardiovascular events, nonsteroidal anti-inflammatory drugs, risk factors

## Abstract

Background: Potential risk factors for major adverse cardiovascular events (MACE) in patients with ankylosing spondylitis (AS) requiring medical therapy should be investigated. Methods: We identified newly diagnosed AS patients without previous MACE from 2004 to 2012 using the National Health Insurance Research Database, matched MACE cases with non-MACE controls at a 1:4 ratio for age, gender, AS duration, and index date, and included 947 AS patients with MACE and 3896 matched controls for final analyses. By using conditional logistic regression analyses, we examined the associations of MACE with low income, urbanisation, comorbidities, common extra-articular manifestations (EAM), and medications, including nonsteroidal anti-inflammatory drugs (NSAID) of three categories (traditional NSAIDs, selective cyclooxygenase-2 inhibitors (COX-2i), and preferential COX-2is) with their annual cumulative defined daily dose (cDDD) within a year before MACE development. Results: MACE development was associated with the use of selective COX-2is (especially with annual cDDD > 132) and corticosteroids, residence in rural regions, and well-known associated comorbidities, but not with the use of traditional NSAIDs, preferential COX-2i, biologics, methotrexate, sulfasalazine, and common EAMs. Conclusions: The risk factors of MACE in newly diagnosed AS patients include residence in rural regions, well-known associated comorbidities, and the use of corticosteroids and selective COX-2is. A major limitation was the lack of information on individual lifestyle patterns and disease activity.

## 1. Introduction

Ankylosing spondylitis (AS), currently also categorised as radiographic axial spondyloarthritis, is a chronic, systemic disease characterised by spondyloarthritis, symmetrical sacroiliitis, peripheral oligoarthritis at the lower limbs, and enthesitis. In Taiwan, AS exhibits male predominance, with a male-to-female ratio of 2.5–3:1 [1], a decreasing prevalence from 0.38% in 1994 [2] to 0.24% in 2010, and an incidence of 0.42–0.50 per 1000 person-years (2005–2010) [3]. AS in Taiwan is diagnosed in accordance with the details in the 1984 modified New York criteria for the classification of AS [4] or the 2009 Assessment of Spondyloarthritis International Society classification criteria for axial spondyloarthritis [5,6]. We followed the consensus recommendations in Taiwan with respect to the management of axial spondyloarthritis [7]. The recommended pharmacologic management includes the use of long-term nonsteroidal anti-inflammatory drugs (NSAIDs), particularly selective cyclooxygenase-2 inhibitors (COX-2i), and the conditional use of local glucocorticoids or conventional synthetic disease-modifying antirheumatic drugs. Some biological and targeted synthetic disease-modifying antirheumatic drugs have been indicated for persistently active or advanced AS despite conventional treatments.

Patients with AS are notably at risk for comorbidities involving the cardiovascular system [8]. They are predisposed to non-ischaemic cardiac involvements, such as conduction abnormalities and arrhythmias [9], valvular and aortic root disease [10], congestive heart failure, and ventricular dysfunction [11]. Studies in Taiwan [12] and other countries worldwide [13] reported that patients with AS are also at risk for various cardiovascular and cerebrovascular diseases. Aside from associated comorbidities (hypertension [12], type 2 diabetes mellitus [14], dyslipidaemia [15], obesity (body mass index ≥ 30 kg/m^2^) [16], etc.), such risk is also due to chronic systemic inflammation [17]. Such extensive cardiovascular involvement or complications of AS are significant public health issues that were easily ignored in previous clinical practice. Accordingly, the development of a predictive chart score [18] or algorithm for the risk of major adverse cardiovascular events (MACE) in patients with AS based on identified risk factors is important for improving holistic management. Epidemiologic studies establishing risk factors in the demographic, socioeconomic, clinical, immunologic, or pharmacologic domain remain lacking to date. Nevertheless, the associations between these treatments and MACE in patients with AS remain unclear. Accordingly, the present study aimed to investigate the factors associated with MACE in patients with incident AS requiring pharmacologic therapy.

## 2. Materials and Methods

### 2.1. Study Design

We conducted a nationwide, population-based, retrospective case–control study using the 2003–2013 Taiwan National Health Insurance Research Database.

### 2.2. Source of Data and Study Population

The claims data of AS patients were obtained from the research database from 1 January 2003 to 31 December 2013. Newly diagnosed AS patients aged ≥20 years were identified if they received an AS diagnosis (the International Codes of Diseases—Ninth Revision—Clinical Modification (ICD-9-CM) code 720.0) at least three times during outpatient visits or at least once during hospitalisation from 1 January 2003 to 31 December 2012. Patients were excluded if they had received an AS diagnosis by the ICD code before 1 January 2004, had a diagnosis of MACE prior to AS, or had missing insurance information from 2003 to 2013. The exclusion of patients with AS diagnosis by the ICD code before 2004 assured the participants were with newly diagnosed AS but a prior history of MACE between 2004 and 2012, and the period between 2004 and 2013 constituted the follow-up period of this study.

### 2.3. Identification of MACE Cases and Non-MACE Matched Controls

MACE was defined as a composite of myocardial infarction, ischaemic stroke, or patients who underwent coronary artery bypass graft or percutaneous coronary intervention [19,20]. Myocardial infarction was identified using ICD-9 code 410.X, except for 410.X2, among inpatients with hospitalisation for at least three days, unless mortality occurred. Ischaemic stroke was identified among inpatients using the hospital discharge diagnosis of codes ICD-9 433–436, except for 433.X0 and 434.X0. For the identification of patients undergoing coronary revascularisation, the ICD-9 procedure codes 00.66, 36.03, 36.06, 36.07, and 36.09 were used for percutaneous coronary intervention, percutaneous transluminal coronary angioplasty, or stent, and ICD-9 procedure codes 36.1 and 36.2, for coronary artery bypass graft.

MACE cases were defined as newly diagnosed AS patients who developed MACE after AS diagnosis, and non-MACE controls were those who did not develop MACE in the follow-up period. All subjects were designated their index dates after enrolment, which was the date of their MACE diagnosis for MACE cases and the date of their first outpatient department visits each year for non-MACE controls. After matching the year of index dates between both groups, we could examine if the occurrence of the variables within a year before MACE development influenced the risk of MACE. We matched MACE cases with non-MACE controls at a ratio of 1:4 for age, gender, AS duration, and year of the index date.

### 2.4. Independent Variables

The independent variables included the baseline characteristics of the patients, such as the age at diagnosis of AS and MACE, gender, urbanisation in accordance with a Taiwanese township stratification model [21], socioeconomic status represented by low income, common comorbidities, three common extra-articular manifestations (EAMs, including acute anterior uveitis, psoriasis, and inflammatory bowel disease), and the use of medications.

Comorbidities by ICD-9 codes included hypertension (ICD-9 codes 401–405), hyperlipidaemia receiving lipid-lowering agents (ICD-9 code 272), diabetes (ICD-9 code 250), chronic kidney disease (CKD, ICD-9 codes 580–587), heart failure (ICD-9 code 428), valvular heart disease (VHD, ICD-9 codes 093.2, 394–397, 424, 746.3–746.6), and chronic obstructive pulmonary disease (COPD, ICD-9 codes 490–493, 496). The comorbidities were identified if the ICD-9 code was documented three times or more during outpatient visits or at least once during hospitalisation.

The EAMs were diagnosed by corresponding specialists and were recognised by ICD-9 codes (acute anterior uveitis, ICD-9 codes 364.00–364.02, 364.04–364.05, and 364.3; psoriasis, ICD-9 code 6961; inflammatory bowel disease, ICD-9 codes 555 and 556). The summary of all included diseases and manifestations and their corresponding ICD-9-CM codes are given in Appendix A.

The medications included antiplatelet and anticoagulation agents; biologics; conventional synthetic disease-modifying antirheumatic drugs, including methotrexate and sulfasalazine; corticosteroids; and NSAIDs. The NSAIDs were further classified into three categories based on their pharmacological COX-2 selectivity, including traditional NSAIDs (e.g., aspirin, ibuprofen, ketorolac, and naproxen), selective COX-2is (e.g., celecoxib and etoricoxib), and preferential COX-2is (e.g., meloxicam and nimesulide) to identify their respective impact on the risk of MACE. The impact on the risk of exposure to a cumulative dosage of NSAIDs within one year before MACE development was explored, so we defined the dosage as the annual cumulative defined daily dose (cDDD) of NSAIDs in general and then broke it into three categories. Four cDDD ranges of each were stratified based on quartiles of the number of NSAIDs users. As the rheumatologists adjusted the dosage of prescribed medications depending on variation in disease status, the annual cDDD could also reflect the yearly medication demand and disease activity.

By using conditional logistic regression analyses, we examined the associations of MACE with the occurrence of the abovementioned variables within a year before MACE development or index date. We also examined the influence of NSAIDs with their annual cDDD on the risk of MACE with a Bonferroni correction for statistical significance.

### 2.5. Patient and Public Involvement

Patients and the public were not involved in the design, conduct, reporting, or dissemination of this research.

### 2.6. Statistical Analysis

The demographic data are shown as mean ± standard deviation for continuous variables and number (percent) for categorical variables. For the comparison of variables between AS participants with MACE and the non-MACE controls, differences in categorical and continuous variables were compared using the χ^2^ test and independent *t*-test, respectively. Conditional logistic regression analyses were conducted to estimate the risks of MACE associated with independent variables, shown as adjusted odds ratios with 95% confidence intervals (CIs) after adjusting for confounders. A probability (*p*) of <0.05 was considered statistically significant and a Bonferroni correction of the probability value cut-off was applied to examine the association between the risk of MACE and three categories of NSAIDs (*p* < 0.0167). Statistical calculations were performed using the Statistical Package for the Social Sciences (SPSS), Windows Version 13.0 (SPSS Inc., Chicago, IL, USA).

### 2.7. Ethics Statement

This study was approved by the Institutional Review Board of Taichung Veterans General Hospital in Taiwan (IRB TCVGH No: CE19038A). The requirement for informed consent from each participant was waived because the used data from Taiwan National Health Insurance Research Database were anonymised.

## 3. Results

### 3.1. Selection and Grouping of Study Population and Prevalence and Incidence of MACE

The flow diagram of enrolment, categorisation, and matching for the comparison of the study population is shown in Figure 1. From 1 January 2003 to 31 December 2012, we identified 63,389 patients with AS. Those with AS as defined by the ICD code before 1 January 2004 (*n* = 18,821) and those who developed MACE before a diagnosis of AS or who had missing insurance information (*n* = 1973) were excluded. A total of 42,595 participants with newly diagnosed AS without prior MACE between 2004 and 2012 were included; of this total number, 1151 patients (prevalence of MACE: 2.7%) developed MACE in the follow-up period, and 41,444 did not. The incidence of MACE in patients with AS was 469.26 per 100,000 person-years in the whole identified population (*n* = 42,595), 483.89 per 100,000 person-years in the male population, and 442.63 per 100,000 person-years in the female population (Appendix A). Ultimately, 974 AS patients with MACE and 3896 (1:4) non-MACE controls, matched for age, gender, AS duration, and year of the index date were included in the final analysis.

### 3.2. Baseline Characteristics of the Study Population

Comparisons of the baseline characteristics surrounding demographics, comorbidities, EAMs and medications between AS patients with and without MACE are shown in Table 1. The mean ages at diagnosis of AS and MACE in the study population were 59 and 63 years, respectively. The study population mainly consisted of male patients (*n* = 3255, 66.8%). The degree of urbanisation (*p* = 0.449) and proportions of patients with a low income (47.9% and 45.6%, respectively, *p* = 0.191) showed no significant difference between the MACE and non-MACE groups. Patients with MACE exhibited high percentages of comorbidities, including hypertension (72.8% vs. 36.3%, *p* < 0.001), hyperlipidaemia receiving lipid-lowering agents (61.3% vs. 19.5%, *p* < 0.001), diabetes (35.8% vs. 14.0%, *p* < 0.001), CKD (9.8% vs. 2.5%, *p* < 0.001), heart failure (11.7% vs. 1.9%, *p* < 0.001), VHD (7.1% vs. 2.0%, *p* < 0.001), and COPD (11.7% vs. 8.4%, *p* = 0.001). The development of MACE was independent of the occurrence of the EAMs. High percentages of patients with MACE received antiplatelet agents (93.7% vs. 21.2%, *p* < 0.001) and anticoagulation agents (46.8% vs. 2.2%, *p* < 0.001). With respect to concomitant medications treating AS, similar percentages of patients in both groups received biologics, methotrexate and sulfasalazine. High percentages of patients with MACE received corticosteroids (48.6% vs. 32.9%, *p* < 0.001) and NSAIDs (91.4% vs. 86.6%, *p* < 0.001). Patients with MACE also consumed high dosages of NSAIDs, as suggested by the high percentages of NSAIDs users (91.4% vs. 86.6%, *p* < 0.001) and high mean cDDD (97.5 vs. 79.7, *p* < 0.001). The results are consistent in users of traditional NSAIDs (percentages: 83.7% vs. 76.3%, *p* < 0.001; mean cDDD: 42.3 vs. 33.5, *p* < 0.001) and selective COX-2is (percentages: 32.1% vs. 26.0%, *p* < 0.001; mean cDDD: 32.9 vs. 26.6, *p* = 0.028), but not in users of preferential COX-2is.

### 3.3. Risk Factors of MACE Development in Patients with AS

After the conditional multivariable logistic regression analyses, the risk of MACE numerically increased in users of NSAIDs in general but significantly increased in users of traditional NSAIDs and selective COX-2is (Table 2). The results are consistent only for users of selective COX-2is after Bonferroni correction. The use of preferential COX-2is showed no such association. The risk was especially associated with the use of traditional NSAIDs with cDDD 7.75–21 and selective COX-2is with cDDD ≤ 28 and >132 (Table 3). The risk was associated with the use of corticosteroids, with a dose–response relationship (Table 2). The risk factors also included residence in rural regions and well-known associated comorbidities, including hypertension, hyperlipidaemia, diabetes, CKD, heart failure, and VHD. Otherwise, low income, EAMs, COPD, and the use of methotrexate, sulfasalazine, and biologics were not associated with MACE development.

## 4. Discussion

This study demonstrated that for newly diagnosed adult AS patients without prior history of MACE, aside from well-known risk factors such as associated comorbidities and the use of corticosteroids, MACE was associated with residence in rural regions but not with the three EAMs. Regarding medications, MACE was associated with the use of selective COX-2is, especially with annual cDDD > 132. The use of traditional NSAIDs, preferential COX-2is, and biologics was associated with no evidence of significant risk. Our study presented the factors associated with the risk of MACE in patients with AS in the Eastern Asian population, which might also constitute a significant issue in public health.

The high risk of MACE in AS patients residing in rural regions was consistent with the findings of studies involving the general population. The high risk of heart disease among nonmetropolitan residents might be attributed to poverty, distance from medical institutions, and limited access to health resources [22,23]. A higher risk of MACE in rural AS patients has not been sufficiently reported in the current English literature. Rural AS patients were found to be older, have labour-intensive jobs, and experienced presenteeism and impaired work productivity [24]. The abovementioned characteristics might make them more prone to MACE development compared with their urban counterparts. The results highlight the need for more health resources involving clinical rheumatology practice in rural regions.

A high prevalence of comorbidities as traditional cardiovascular risk factors was found among the AS population [12], presenting an atherogenic metabolic profile that leads to subclinical atherosclerosis [25]. A few studies presented inconsistent results and thus indicated other contributing factors aside from the traditional risk factors [26,27]. Systemic inflammation also potentiates the occurrence of MACE in the AS population [28], and no elevated risk was reported for those with low disease activity and that are free from concomitant comorbidities [29]. Despite the increased prevalence of CKD [30], heart failure or ventricular dysfunction [11,13], VHD [10,31], and COPD [32], little is known about the independent causal relationship between these organ dysfunctions and the risk of MACE. The associations might be partly explained by associated factors [33] and interrelated pathophysiology [34,35].

No EAMs included in this study were associated with the risk of MACE. Currently, no study in the English literature has established the relationship between EAMs of AS and the risk of MACE. EAMs were regarded as the effects of uncontrolled systemic inflammation [36]. Thus, our result might be biased by the disease activity and medication profile of each participant that could not be acquired in our study.

The most important finding in this study was the association of the risk of MACE with the use of selective COX-2is, especially in those with cDDD > 132. The use of traditional NSAIDs and preferential COX-2is showed no evidence of association with the risk of MACE (Table 2). Selective COX-2is were frequently used among Taiwanese AS patients, and those with such high cDDD were more likely to have persistently active AS. The result might thus reflect their high disease activity, yet it still signified that AS patients would face an increased risk of MACE even with adequate inflammation control using NSAIDs. Despite the preferred use of NSAIDs for AS in current practice for fear of gastrointestinal adverse effects, an excess dosage of selective COX-2is for disease control might confer higher cardiovascular risk. Selective COX-2is would particularly inhibit the synthesis of prostacyclin by endothelial cells, disrupt the homeostasis between thromboxane A2 and prostacyclin levels, and result in thrombosis [37,38]. This finding illuminated the importance of avoiding excess dosage; in addition, preferential COX-2is, such as meloxicam, can be considered as an alternative medication for AS patients at risk for MACE, with concerns about cardiovascular safety.

The appropriate use of NSAIDs might confer cardiovascular protection for AS patients [39,40] when compared with the general population [41]. We nevertheless reported the opposite results concerning the increased risk of MACE, specifically with the use of selective COX-2is. Control of systemic inflammation was the keystone of cardioprotection for AS patients [17]. However, the inadequate use of NSAIDs could still increase cardiovascular toxicity, as Braun et al. suggested elevated cardiovascular risk in long-term NSAID users with improper indications [40]. The significantly higher risk among selective COX-2is users with annual cDDD > 132 signified the harm of excess use. Our results conflict with those in previous Taiwanese case–control studies, as Wu et al. noted a decreased risk of coronary artery disease among celecoxib users with an average DDD > 1.5 (>300 mg) [42], and Tsai et al. revealed a protective effect of long-term frequent NSAID use [43]. By comparison, our study focused on the composite outcome of MACE and analysed the influence of three pharmacological categories of NSAIDs on the risk of MACE with more participants. We represented the annual dosage as cDDD within a year, which helped us to adjust the confounding effect of compliance with outpatient follow-up and disease activity, considering that a high dosage was prescribed for active AS in real-world practice. Aside from traditional NSAIDs, we also identified no impact of preferential COX-2is with annual cDDD on the risk of MACE in the AS population. In the general population, a nationwide case–control study revealed a modest association of first MI with the current use of preferential COX-2is [41]. This study, together with others [44,45], indicated that meloxicam, the most commonly used preferential COX-2i, had no elevated cardiovascular toxicity compared with other NSAIDs.

Some studies supported the cardioprotective effect of disease-modifying antirheumatic drugs, including sulfasalazine [42] and tumour necrosis factor-alpha inhibitors, on patients with AS. Though tumour necrosis factor-alpha inhibitors were associated with heart failure, it was not absolutely contraindicated for those with mild heart failure, provided cautious patient selection and follow-up [46]. Tumour necrosis factor-alpha inhibitors diminished complement activation [47], systemic inflammation, and subclinical atherosclerosis, followed by the modulation of lipid profiles [48]. However, whether the above changes confer a clinically cardiovascular benefit in AS remains unclear [39]. Despite limited data, interleukin-17A inhibitors are anticipated to exhibit atheroprotection owing to their anti-inflammatory effects [49]. Although biologics have not been widely used in Taiwan, we still found a non-significant decrease in the risk of MACE with the use of biologics in patients with AS, implying its cardioprotective potential, which may be expected in future trials with more users and longer observational periods.

This study had several limitations. Firstly, the results of the retrospective case–control study had limited strength. The use of an administrative database could prevent us from obtaining information on important known risk factors, including disease activity, cardiac autonomic function [50], and lifestyle patterns such as smoking status, nutritional status, dietary habits, and stressful or psychiatric problems. Laboratory information involving plasma homocysteine [51] concentration was inaccessible. Additionally, real compliance with medications and outpatient follow-up and concomitant alternative medication use remained unclear. These all could be prominent confounders. Secondly, the inclusion of subjects via the identification of ICD codes could not avoid selection bias, and the accuracy of the diagnosis might also be a concern, despite the inclusion criteria of the ICD code of AS being issued by rheumatologists at least three times during outpatient visits or at least once during hospitalisation. Thirdly, as a population-based study in Taiwan, generalisability may be limited for other regions in the world. Facing the abovementioned limitations, we adjusted the potential confounders, such as age, gender, and AS duration, with the use of conditional multivariable analyses. We used the cDDD of prescribed NSAIDs to simultaneously represent the medication demand, degree of inflammation, and disease activity. It could also positively approximate the compliance with outpatient follow-up. In this way, the adjustment of the influence of cDDD could thus partially adjust the confounding effect of disease activity and non-compliance.

## 5. Conclusions

This nationwide, population-based, case–control study revealed the risk factors of MACE in patients with newly diagnosed AS, including residence in rural regions, well-known associated comorbidities, and the use of corticosteroids and selective COX-2is. The major limitation of this study was a lack of direct information on individual lifestyle patterns and disease activity. This suggested avoiding the use of selective COX-2is for improper indications and with excess dosages for AS patients at risk for MACE, which spotlighted the adverse effects of NSAIDs and might aid in the development of a predictive chart score or algorithm for the risk of MACE in patients with AS.

## Figures and Tables

**Figure 1 ijerph-19-04098-f001:**
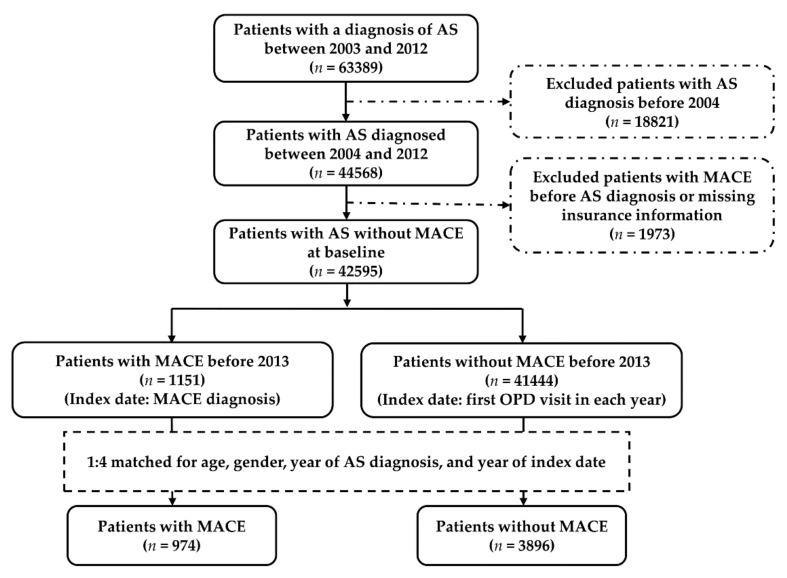
Flow diagram of enrolment, categorisation, and matching for comparison of the study population. AS, ankylosing spondylitis. MACE, major adverse cardiovascular event. OPD, outpatient department.

**Table 1 ijerph-19-04098-t001:** Baseline characteristics of matched study patients with and without MACE.

	Non-MACE (*n* = 3896)	MACE (*n* = 974)	*p* Value
Age at diagnosis of AS, years	59.3 ± 12.1	59.3 ± 12.1	1.000
Age at diagnosis of MACE, years	63.1 ± 12.0	63.1 ± 12.0	1.000
Male, *n* (%)	2604 (66.8)	651 (66.8)	1.000
Urbanisation, *n* (%)			0.449
Urban	1035 (26.6)	240 (24.6)	
Suburban	1647 (42.3)	418 (42.9)	
Rural	1213 (31.1)	316 (32.4)	
Low income, *n* (%) ^1^	1777 (45.6)	467 (47.9)	0.191
Comorbidities			
Hypertension, *n* (%)	1415 (36.3)	709 (72.8)	<0.001
Hyperlipidaemia receiving lipid-lowering agents, *n* (%)	761 (19.5)	619 (61.3)	<0.001
Diabetes mellitus, *n* (%)	546 (14.0)	349 (35.8)	<0.001
Chronic kidney disease, *n* (%)	96 (2.5)	95 (9.8)	<0.001
Heart failure, *n* (%)	75 (1.9)	114 (11.7)	<0.001
Valvular heart disease, *n* (%)	77 (2.0)	69 (7.1)	<0.001
COPD, *n* (%)	327 (8.4)	114 (11.7)	0.001
Extra-articular manifestations			
Acute anterior uveitis, *n* (%)	321 (8.2)	80 (8.2)	0.979
Psoriasis, *n* (%)	91 (2.3)	25 (2.6)	0.672
Inflammatory bowel disease, *n* (%)	23 (0.6)	4 (0.4)	0.499
Antiplatelet agents, *n* (%)	825 (21.2)	913 (93.7)	<0.001
Anticoagulation agents, *n* (%)	86 (2.2)	456 (46.8)	<0.001
Biologics, *n* (%)	35 (0.9)	7 (0.7)	0.588
Methotrexate, *n* (%)	139 (3.6)	30 (3.1)	0.457
Sulfasalazine, *n* (%)	676 (17.4)	165 (16.9)	0.762
Steroid use, *n* (%)	1282 (32.9)	473 (48.6)	<0.001
No use	2614 (67.1)	501 (51.4)	
<5 mg/day	1081 (27.7)	296 (30.4)	
≥5 mg/day	201 (5.2)	177 (18.2)	
NSAIDs, *n* (%)	3372 (86.6)	890 (91.4)	<0.001
NSAIDs, cDDD	79.7 ± 104.4	97.5 ± 117.5	<0.001
None	524 (13.4)	86 (8.6)	<0.001
0 < cDDD ≤ 18.75	856 (22.0)	213 (21.9)	
18.75 < cDDD ≤ 56	872 (22.4)	197 (20.2)	
56 < cDDD ≤ 131.25	831 (21.3)	228 (23.4)	
131.25 < cDDD	813 (20.9)	252 (25.9)	
Traditional NSAIDs, *n* (%)	2974 (76.3)	815 (83.7)	<0.001
Traditional NSAIDs, cDDD	33.5 ± 58.0	42.3 ± 68.2	<0.001
None	922 (23.7)	159 (16.3)	<0.001
0 < cDDD ≤ 7.75	768 (19.7)	188 (19.3)	
7.75 < cDDD ≤ 21	759 (19.5)	209 (21.5)	
21 < cDDD ≤ 56.17	726 (18.6)	192 (19.7)	
56.17 < cDDD	721 (18.5)	226 (23.2)	
Selective COX-2 inhibitors, *n* (%)	1014 (26.0)	313 (32.1)	<0.001
Selective COX-2 inhibitors, cDDD	26.6 ± 75.3	32.9 ± 80.2	0.028
None	2882 (74.0)	661 (67.9)	0.005
0 < cDDD ≤ 28	327 (8.4)	104 (10.7)	
28 < cDDD ≤ 56	185 (4.7)	56 (5.7)	
56 < cDDD ≤ 132	250 (6.4)	74 (7.6)	
132 < cDDD	252 (6.5)	79 (8.1)	
Preferential COX-2 inhibitors, *n* (%)	1339 (34.4)	355 (36.4)	0.223
Preferential COX-2 inhibitors, cDDD	19.6 ± 48.3	22.3 ± 52.3	0.142
None	2557 (65.6)	619 (63.6)	0.645
0 < cDDD ≤ 14	390 (10.0)	99 (10.2)	
14 < cDDD ≤ 30	284 (7.3)	80 (8.2)	
30 < cDDD ≤ 77	343 (8.8)	85 (8.7)	
77 < cDDD	322 (8.3)	91 (9.3)	

Values are mean ± standard deviation (S.D.) or %. ^1^ Receiving ≤ 21,000 TWD per month. AS, ankylosing spondylitis. cDDD, cumulative defined daily dose. COPD, chronic obstructive pulmonary disease. COX-2, cyclooxygenase-2. MACE, major adverse cardiovascular event. NSAIDs, nonsteroidal anti-inflammatory drugs.

**Table 2 ijerph-19-04098-t002:** Factors associated with MACE in patients with ankylosing spondylitis.

	Univariable Analysis		Multivariable Analysis (Model 1—Adjustment of NSAIDs in General)		Multivariable Analysis (Model 2—Adjustment of Three NSAIDs Categories)	
Independent Variable: MACE	Odds Ratio (95% CI)	*p*	Odds Ratio (95% CI)	*p*	Odds Ratio (95% CI)	*p*
Urbanisation						
Urban	1 (Reference)		1 (Reference)		1 (Reference)	
Suburban	1.09 (0.92–1.30)	0.315	1.22 (0.98–1.52)	0.076	1.22 (0.98–1.52)	0.082
Rural	1.13 (0.93–1.36)	0.218	1.32 (1.03–1.69)	0.026	1.32 (1.03–1.69)	0.028
Low income ^1^	1.17 (1.01–1.35)	0.041	1.14 (0.94–1.38)	0.177	1.13 (0.94–1.37)	0.206
Comorbidities						
Hypertension	5.42 (4.58–6.43)	<0.001	3.12 (2.57–3.80)	<0.001	3.12 (2.57–3.80)	<0.001
Hyperlipidaemia	7.45 (6.31–8.78)	<0.001	4.93 (4.09–5.95)	<0.001	5.00 (4.14–6.03)	<0.001
Diabetes mellitus	3.63 (3.07–4.29)	<0.001	1.71 (1.39–2.09)	<0.001	1.69 (1.37–2.07)	<0.001
Chronic kidney disease	4.29 (3.19–5.77)	<0.001	1.97 (1.34–2.88)	0.001	1.98 (1.35–2.90)	0.001
Heart failure	6.77 (4.98–9.20)	<0.001	4.04 (2.74–5.94)	<0.001	4.04 (2.74–5.94)	<0.001
Valvular heart disease	3.91 (2.78–5.50)	<0.001	2.10 (1.36–3.26)	0.001	2.06 (1.33–3.20)	0.001
COPD	1.47 (1.16–1.84)	0.001	0.92 (0.68–1.23)	0.565	0.92 (0.68–1.23)	0.563
Extra-articular manifestations						
Acute anterior uveitis	1.00 (0.77–1.29)	0.979	1.03 (0.74–1.44)	0.849	1.01 (0.73–1.41)	0.952
Psoriasis	1.10 (0.70–1.74)	0.669	1.10 (0.61–1.98)	0.750	1.13 (0.62–2.04)	0.694
Inflammatory bowel disease	0.70 (0.24–2.01)	0.503	0.82 (0.23–2.95)	0.757	0.77 (0.21–2.83)	0.694
Biologics	0.80 (0.36–1.80)	0.590	0.41 (0.14–1.20)	0.103	0.37 (0.13–1.11)	0.076
Methotrexate	0.86 (0.57–1.28)	0.454	0.72 (0.42–1.26)	0.250	0.70 (0.40–1.22)	0.204
Sulfasalazine	0.97 (0.80–1.18)	0.750	0.93 (0.72–1.19)	0.553	0.87 (0.67–1.12)	0.278
Steroid						
None	1 (Reference)		1 (Reference)		1 (Reference)	
<5 mg/day	1.45 (1.23–1.70)	<0.001	1.29 (1.06–1.58)	0.013	1.25 (1.02–1.54)	0.028
≥5 mg/day	4.80 (3.81–6.06)	<0.001	4.85 (3.58–6.55)	<0.001	4.75 (3.51–6.43)	<0.001
NSAIDs	1.67 (1.31–2.13)	<0.001	1.27 (0.94–1.71)	0.118		
Traditional NSAIDs	1.61 (1.34–1.95)	<0.001			1.29 (1.02–1.63)	0.031
Selective COX-2 inhibitors	1.37 (1.17–1.60)	<0.001			1.38 (1.13–1.69)	0.002
Preferential COX-2 inhibitors	1.11 (0.95–1.29)	0.198			0.91 (0.75–1.10)	0.320

^1^ Receiving ≤ 21,000 TWD per month. COPD, chronic obstructive pulmonary disease. COX-2, cyclooxygenase-2. MACE, major adverse cardiovascular event. NSAIDs, nonsteroidal anti-inflammatory drugs.

**Table 3 ijerph-19-04098-t003:** Associations between annual cumulative dosage of NSAIDs and MACE in patients with ankylosing spondylitis.

	Univariable Analysis		Multivariable Analysis (Model 1—Adjustment of NSAIDs in General)		Multivariable Analysis (Model 2—Adjustment of Three NSAIDs Categories)	
Independent Variable: MACE	Odds Ratio (95% CI)	*p*	Odds Ratio (95% CI)	*p*	Odds Ratio (95% CI)	*p*
NSAIDs	1.67 (1.31–2.13)	<0.001				
None	1 (Reference)		1 (Reference)			
0 < cDDD ≤ 18.75	1.56 (1.18–2.05)	0.002	1.33 (0.96–1.85)	0.090		
18.75 < cDDD ≤ 56	1.44 (1.09–1.91)	0.011	1.13 (0.81–1.58)	0.480		
56 < cDDD ≤ 131.25	1.77 (1.34–2.34)	<0.001	1.25 (0.89–1.76)	0.199		
131.25 < cDDD	2.00 (1.52–2.64)	<0.001	1.38 (0.97–1.95)	0.072		
Traditional NSAIDs	1.61 (1.34–1.95)	<0.001				
None	1 (Reference)				1 (Reference)	
0 < cDDD ≤ 7.75	1.43 (1.13–1.80)	0.003			1.28 (0.96–1.70)	0.095
7.75 < cDDD ≤ 21	1.63 (1.29–2.05)	<0.001			1.39 (1.05–1.85)	0.023
21 < cDDD ≤ 56.17	1.57 (1.24–1.99)	<0.001			1.25 (0.93–1.67)	0.140
56.17 < cDDD	1.87 (1.49–2.36)	<0.001			1.28 (0.96–1.72)	0.095
Selective COX-2 inhibitors	1.37 (1.17–1.60)	<0.001				
None	1 (Reference)				1 (Reference)	
0< cDDD ≤ 28	1.41 (1.11–1.80)	0.005			1.37 (1.02–1.85)	0.036
28 < cDDD ≤ 56	1.34 (0.98–1.83)	0.064			1.43 (0.96–2.12)	0.077
56 < cDDD ≤ 132	1.32 (0.99–1.73)	0.051			1.20 (0.85–1.71)	0.300
132 < cDDD	1.38 (1.06–1.81)	0.019			1.61 (1.12–2.32)	0.011
Preferential COX-2 inhibitors	1.11 (0.95–1.29)	0.198				
None	1 (Reference)				1 (Reference)	
0 < cDDD ≤ 14	1.06 (0.83–1.35)	0.654			0.83 (0.62–1.11)	0.213
14 < cDDD ≤ 30	1.17 (0.90–1.53)	0.237			1.00 (0.71–1.40)	0.996
30 < cDDD ≤ 77	1.04 (0.80–1.35)	0.778			0.88 (0.64–1.22)	0.445
77 < cDDD	1.18 (0.92–1.53)	0.198			0.98 (0.71–1.35)	0.899

cDDD, cumulative defined daily dose. CI, confidence interval. COX-2, cyclooxygenase-2. MACE, major adverse cardiovascular event. NSAIDs, nonsteroidal anti-inflammatory drugs.

## Data Availability

All the data relevant to the study are included in the article or uploaded as Appendix A.

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
