# Peer review of "Factors Associated with the Risk of Major Adverse Cardiovascular Events in Patients with Ankylosing Spondylitis: A Nationwide, Population-Based Case—Control Study"

_ijerph, 2022, doi:10.3390/ijerph19074098_

Round 1

Reviewer 1 Report

The present article written by Kao CM et al., regarding the “Factors Associated with the Risk of Major Adverse Cardiovascular Events in Patients with Ankylosing Spondylitis: a Nation-wide, Population-based Case-Control Study” is important for the research filed in order to have a better overview of the proper pharmacotherapy, despite several limitations of the present study (already presented in the article).

However, I have some suggestions that might improve the present manuscript:

  1. In the Materials and Methods part, please add more information regarding the NSAIDs that were prescribed (which substances) referring also to the subgroup of the NSAID (traditional, … etc).
  2. More information regarding why/how you chose cDDD intervals would be also helpful for the reader. (cDDD intervals for each subgroup of NSAIDs)
  3. I find the Discussion part to be hard to read. I my opinion there is a lot of information, but too concentrated. Please explain better the main findings of the present article, what is the possible explanation that underlines them and how they correlate with scientific literature.
  4. In the Conclusion part you mentioned “the importance of selecting proper NSAIDs and avoiding excess dosage for AS patients at risk for MACE”. This statement should be better emphasized and explained in the Discussion part, so that the reader understands better the conclusion.
  5. Some parts of the present manuscript require English editing.

Author Response

Thanks for your precious feedback and advice. These are indeed the points that can be improved, and we had made improvements as possible as we could. The response was submitted as the Word file.

Reviewer 2 Report

Thank you for giving us the opportunity to review your manuscript titled "Factors Associated with the Risk of Major Adverse Cardiovascular Events in Patients with Ankylosing Spondylitis: a Nationwide, Population-based Case-Control Study". This manuscript is a very significant contribution to scholarship on adverse events (pharmacoepidemiology) and ankylosing spondylitis (public health), well researched, well organized, and well written. It will be tremendously helpful to all of us who work in this field, and I anticipate it will be widely read and cited. IJERPH should absolutely publish it.

The paper is rich in acronyms, almost all of which I was unfamiliar and thus almost immediately forgot once defined. I would suggest replacing most, if not all, with full names.

On the other hand, this reviewer suggests including a table to describe all together of the International Codes of Diseases–Ninth Revision Clinical Modification (ICD-9-CM) codes (page 3 of 16) instead of describing them intratext.

Please, standardize and clarify the time framework:

  • Abstract section (page 1, line 43): "from 2004 to 2012".
  • Materials and methods section (page 2, lines 103-104): "We conducted a nationwide, population-based, retrospective case-control study using the 2003–2013 Taiwan National Health Insurance Research Database (NHIRD)."
  • Line 107: "...from January 1, 2003, to December 31, 2013...".
  • Page 4, lines 111-113: "...from 2003 to 2012. Patients were excluded if they were diagnosed with AS before January 1, 2004, had a diagnosis of MACE prior to AS, or missed insurance information from 2003 to 2013."
  • Results section, page 170-176: "From January 1, 2003, to December 31, 2013, we identified 63,389 patients with AS from 2004 to 2012. Those who were diagnosed with AS before January 1, 2004 (n = 18821) and those who developed MACE before a diagnosis of AS or who missed the insurance information (n = 1973) were excluded. A total of 42,595 participants who were diagnosed with AS without prior MACE between 2004 and 2012 were included, and 1151 of them (prevalence of MACE: 2.7%) developed MACE before January 1, 2013...".

It's just a little confusing.

Although  there are both statements on Institutional Review Board and Informed Consent (page "3" of 16 -review this number, it might be page 14-, lines 340-343), it might be also appropriate to include them as a last paragraph on Materials and methods section (line 166, just before Results section).

Author Response

(The authors gave the same response as above.)

Reviewer 3 Report

This article entitled "Factors Associated with the Risk of Major Adverse Cardiovascular Events in Patients with Ankylosing Spondylitis: a Nationwide, Population-based Case-Control Study" investigated an important topic. The investigators showed that some of the medications were associated with the risk of MACEs in patients with AS. There are a few comments need to be addressed before proceeding: 

Abstract

Conclusion: The sentence "we emphasized the safe use of medications in AS patients at risk for MACE" is unclear which medications?. Also, based on the limitations of the study, the conclusion sound needs to be lower.

Introduction: 

The background and the importance of the topic are well addressed. 

Patients and methods/ Results:

If NSAIDs are one of the over-counter drugs, estimation of their use and dose are not accurate. 

The data are missing the clinical information, such as disease activity, physical activity which are important cofounders with the outcome. 

The authors need to explain how did the matching. 

Do you know the indications of steroid in these patients, as we know it is not a regular drug for patients with AS.

The type of biologics used; TNFi or ts DMARDs as some of the new biologics have CVD side effects. 

The 2 right columns in table 2 are confusing and need a better label. 

Table 3 ???

Discussion: 

Well written but need to be wrapped. 

Conclusion: 

With lacking of many of the important risk factors, the sound of the conclusion need to be lower. 

Author Response

Thanks for your precious feedback and advice. These are indeed the points that can be improved. Though some limitations related to the nature of the study design might not be amendable and avoidable, we have made improvements as possible as we could. The response was submitted as the Word file.
